# Fischer–Tropsch Synthesis as the Key for Decentralized Sustainable Kerosene Production †

**Andreas Meurer ***  **and Jürgen Kern**

Department of Energy Systems Analysis, Institute of Networked Energy Systems,
German Aerospace Center (DLR), Curiestraße 4, 70563 Stuttgart, Germany; juergen.kern@dlr.de
* Correspondence: andreas.meurer@dlr.de; Tel.: +49-711-6862-8100
† This paper is an extended version of our conference paper from 15th SDEWES Conference, Cologne, Germany, 1–5 September 2020.

**Abstract:** Synthetic fuels play an important role in the defossilization of future aviation transport. To reduce the ecological impact of remote airports due to the long-range transportation of kerosene, decentralized on-site production of synthetic paraffinic kerosene is applicable, preferably as a near-drop-in fuel or, alternatively, as a blend. One possible solution for such a production of synthetic kerosene is the power-to-liquid process. We describe the basic development of a simplified plant layout addressing the specific challenges of decentralized kerosene production that differs from most of the current approaches for infrastructural well-connected regions. The decisive influence of the Fischer–Tropsch synthesis on the power-to-liquid (PtL) process is shown by means of a steady-state reactor model, which was developed in Python and serves as a basis for the further development of a modular environment able to represent entire process chains. The reactor model is based on reaction kinetics according to the current literature. The effects of adjustments of the main operation parameters on the reactor behavior were evaluated, and the impacts on the up- and downstream processes are described. The results prove the governing influence of the Fischer–Tropsch reactor on the PtL process and show its flexibility regarding the desired product fraction output, which makes it an appropriate solution for decentralized kerosene production.

**Keywords:** alternative fuels; power-to-liquid; synthetic fuel; synthetic kerosene; aviation fuel; sustainable fuel; power-to-x; e-fuel; fischer–tropsch; renewable fuel

## 1. Introduction

### 1.1. Motivation

An annual increase in air traffic of over 4% on average is expected during the next decades [1]. Already today, the aviation sector accounts for around 11% of the energy consumption of the entire transport sector [2] and thus contributes significantly to the global greenhouse gas emissions. Even under the consideration of further technology developments and efficiency improvements, the aviation sector could emit 3 times the current amount of $CO_2$ by 2050 if no actions are taken [3]. Although the aviation sector is comparatively seen as the most difficult to decarbonize as there is no feasible short-term possibility for aircraft electrification [3], sustainable aviation fuels (SAF) based on biogenic raw materials and renewable energy represent an option to significantly decrease the emissions of the aviation sector. However, SAF currently account for only about 0.1% of the total kerosene consumption [2].

Despite the fact that the vast majority of current SAF is represented by biofuels [4], kerosene produced via the power-to-liquid (PtL) process based on renewable electrical energy offers a viable option for a future sustainable aviation fuel supply [5].

### 1.2. Related Studies

There are various studies addressing the topic of SAF with a focus on biofuels (e.g., [6–9]). Mawhood et al. [6] presented possible production routes and evaluated the related technologies based on their future potentials. The role of Fischer–Tropsch (FT) synthesis as part of biofuel production was presented by Ail and Dasappa [7], considering the literature data from FT processes under operation. Hamelinck et al. [10] developed a process model based on Aspen Plus® for the technical and economical evaluation of a Biomass-to-Liquid (BtL) process. Similar studies were performed by Sudiro and Bertucco [11] and Lee et al. [12], providing simulation models for different production routes based on Aspen Plus® with a main focus on gasoline and diesel.

Schmidt et al. [5] introduced the PtL process as a relevant option for aviation fuel production and provided techno-economic and environmental comparisons between different process routes based on the literature data. An extensive simulation model for a PtL process was developed by König et al. [13] with Aspen Plus® providing conclusions regarding the process internal correlations and overall efficiencies. Studies concerning the decentralization of FT-based fuel production were performed by Kirsch et al. [14] providing insights on the current state of technology development by the Institute for Micro Process Engineering of Karlsruhe Institute of Technology and the INERATEC GmbH (Karlsruhe, Germany). The work presented here is an extension of a conference paper [15].

### 1.3. Novelty

The novelty of this work is the description of a process chain tailored to the decentralized and sustainable production of kerosene, which can be used directly on site. In the case of Brazil, the current highly centralized production of kerosene [16] in conjunction with the huge state territory leads to long transportation routes across many state borders and, therefore, results in high kerosene prices for remote airports. A decentralized production of kerosene on site might, therefore, already be cost competitive and represent a viable option as an early-stage application.

This paper also describes the development of a Fischer–Tropsch reactor model as part of a future open source process simulation model based on Python, showing the importance of the FT reactor as the core of the PtL process. The purpose of the modular Python-based process model framework is to support the trend towards open and linkable integrated models and to facilitate the system analytical assessment of different fuel production pathways by enabling the possibility of a direct coupling with an energy system or scenario assessment models. This will further ease the multi-criteria assessment and optimization in conjunction with, e.g., open life-cycle-assessment tools. The targeted technical level of detail is, therefore, lower than that made possible by commercial software, such as, e.g., Aspen Plus®, but enables a sufficient representation of the process-related main degrees of freedom.

### 1.4. Summary

Under the consideration of general assumptions regarding future synthetic fuel certification, we qualitatively compared various possibilities for the synthetic generation of kerosene, and the most suitable production pathway for a decentralized application in remote areas was determined to be the PtL process. The process route was examined subdivided into its three main sections, namely synthesis gas generation, synthetic crude production, and crude refining, considering the current technologies. For each main process step, the currently relevant technical possibilities are described, and the specific technologies fitted to the desired product, the decentralized application, and the entire process chain were determined, showing that the process including a synthesis via a FT reaction was the most advantageous.

For demonstration of the influence of the Fischer–Tropsch reactor as key part of the process, the development of the Python-based reactor model is described. The impacts of the main process parameters are shown by means of the model, and the resulting effects

on the up- and downstream processes are pointed out, showing the reactor temperature to be the most prominent operation parameter for a targeted syncrude composition and, thus, the decisive character of the reactor.

## 2. Materials and Methods

### 2.1. Assumptions and Limitations

At the present time, the technical specifications to be met of the main fuel types used in civil aviation—Jet A and Jet A-1—are defined by the international standards ASTM D1655 [17] and DEF STAN 91–91 [18]. The consideration of SAF and definition of its requirements is regulated by annexes of the ASTM D7566 [19] providing various approved production pathways.

One of the seven currently approved production routes describes a Synthetic Paraffinic Kerosene (SPK) via a Fischer–Tropsch reaction (FT-SPK) [19]. To be certified as a drop-in fuel according to ASTM D7566, the FT-SPK may only be used as a blend with conventional jet fuel from crude oil with a maximum blending ratio of up to 50%. Currently, even among the other certified production routes, there is no short-term possibility for the production of a certified Jet A or Jet A-1 with a sustainable kerosene share of more than 50%.

As one of the main drivers behind the idea of sustainable decentralized kerosene production is the avoidance of long-distance transport of fuel from the refinery to the consumer, two main assumptions were made regarding the background of the general idea of the presented application.

#### 2.1.1. Certification of 100% FT-SPK

This study treats the use of 100% FT-SPK as near-drop-in fuel. The underlying assumption is that the use of 100% FT-SPK will be certified for use in slightly modified aircrafts (e.g., compatible sealings [20,21]). In contrast, drop-in fuels have to be compatible with the whole legacy fleet.

#### 2.1.2. Quality Testing

The second main assumption was made regarding the jet fuel quality testing process itself. Currently, every batch produced must pass a series of quality tests in certified laboratories before it can be released for use as fuel for civil aviation [17,22]. A decentralized production, especially in remote regions of comparatively small quantities would—if this test procedure were to be maintained at the same level—involve immense logistical and financial expenses and might not be sustainable and feasible. Accordingly, we assumed that adapted regulations and procedures will be developed for a sustainable decentralized kerosene production in the future, which will enable certified quality testing on site.

### 2.2. Production Route

The production of alternative fuels is possible through many different process routes that can be categorized in different ways. One type of categorization is based on the underlying feedstock, subdividing them into the biomass-to-liquid process—covering the production of "biofuels" based on biogenic raw materials of different types and the power-to-liquid process, shown in Figure 1, which describes a synthetic production based solely on electrical energy as energy input—providing "e-fuels". These variants can be supplemented by two further minor options, a mix of both previous types, the power–biomass-to-liquid (PBtL) and the technology defined as the sun-to-liquid process (StL), which uses sunlight for the direct production of synthesis gas either in a photo-electrochemical cell or via a thermochemical reactor [23].

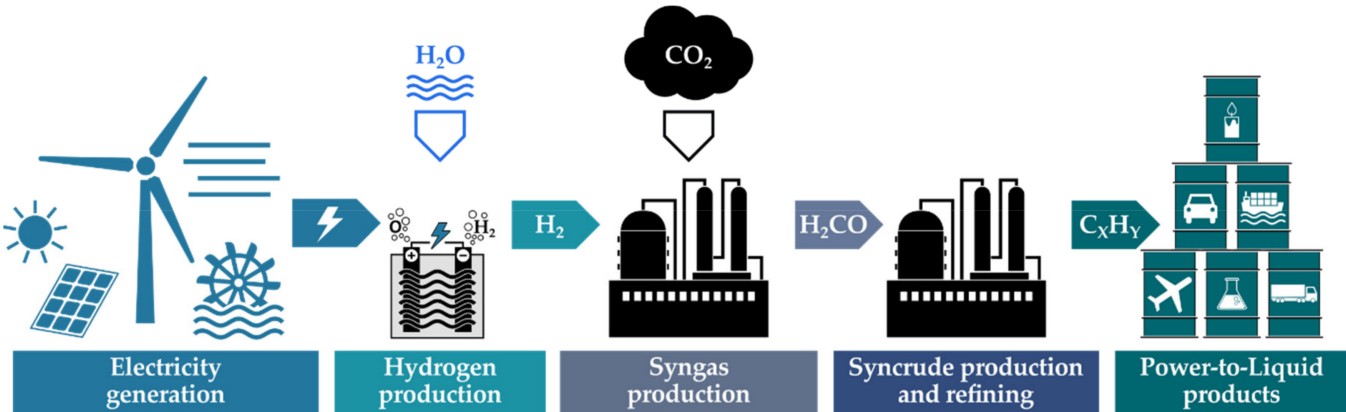

**Figure 1.** Generic scheme of a power-to-liquid process.

Even though the latter StL technology path may be an interesting option in the future, it will not be considered in the further course of the work as it is still in the stage of development [24,25].

For a classification of the other above-mentioned production routes with regard to their suitability for a decentralized and sustainable application, it is necessary to evaluate the role of the required process-specific feedstocks in particular.

### 2.2.1. Feedstock Availability

A decentralization of kerosene production as presented in this work not only includes the final products decoupling of central generation and transport structures but also the localization of the upstream raw material and educts supply. As there is no possibility to compensate for product overcapacities or bottlenecks via connected infrastructure, on-site matching between the demand and supply is crucial. For operational and economic reasons, it is still relevant to avoid unnecessary oversizing of the plant and to strive for the most constant plant operation possible.

As a result, a constant supply of raw materials is necessary. If one compares the two mentioned production categories in this respect, there are clear advantages for the variant of an electrical energy supply for the following reasons. For a sustainable use of biomass, it must be ensured that only residues from other biomass processing sectors are used or that the targeted fuel-related biomass production meets the current requirements of sustainability criteria [26–29].

In the case of a remote application, the probability that an industry will have usable biomass available as a by-product or residue produced at a consistent quality and sufficient quantity is low, even without considering the necessity of year-round availability.

Electrical energy as feedstock offers greater flexibility in this respect, as it is not dependent on local structures or the raw material supply of third parties. Decentralized electrical energy generation using a mix of photovoltaics, wind, and water power tailored to the location, its energy potentials and the plants demand can provide the required energy. Short-term fluctuations can be compensated via electrical energy storages or by buffering them through hydrogen storage within the process.

### 2.2.2. Local Impacts

To evaluate the above-mentioned possibility of using certified biomass as raw material, the main local environmental influences were compared in the following. The key aspects to consider were the specific water and land demands related to the amount of fuel produced. Both the water and land demands of biomass-based fuels are highly dependent on the specific types of feedstock and vary between approximately 500 to 20,000 $L_{water}/L_{fuel}$ and 0.85 to 17.3 $m^2/L_{fuel}/y$ respectively. The comparison with the parameters of a PtL production that result in a water demand of up to 1.38 $L_{water}/L_{fuel}$ and a land demand

between 0.33 and 0.74 m$^2$/L$_{fuel}$/y [25] (varying due to the type of energy production) showed the local disadvantages of a production based on primary biomass.

To summarize—under certain conditions, especially if usable biogenic residues are available as raw materials in sufficient consistency, quality, and quantity or if there are potentials for sustainable use of primary biomass, the use of a BtL process is reasonable. However, as this cannot be presupposed for the decentralized application under consideration here, PtL is clearly defined as the preferred process route and will be examined subsequently.

*2.3. Process Design*

The simplified process design outlined in the following section is based on the idea of a design with a clear focus on the final on-specification product kerosene considering the above production route selection and requirements, in particular the avoidance of biogenic raw materials and the exclusive use of renewable electrical energy as an external energy source. An exemplary flow sheet of the simplified resulting process setup following [13] is shown in Figure 2.

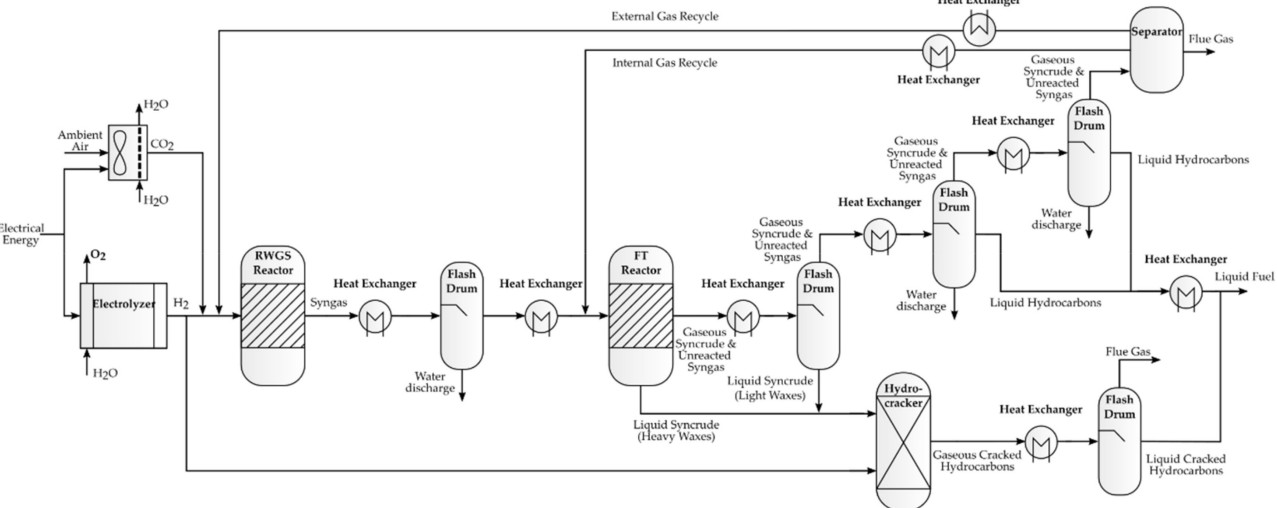

**Figure 2.** A simplified process flow sheet of a possible power-to-liquid process for a decentralized application with a focus on kerosene as the main product.

To decide on a specific production route, the suitability of the various possible PtL routes for decentralized application are considered. As part of this category, currently there are mainly two relevant options for the production of liquid hydrocarbons in the kerosene range—the production via methanol synthesis, also described as alcohol-to-jet (AtJ) and the production via Fischer–Tropsch synthesis.

The main difference between the two catalytic synthesis types lies in the synthesis reaction itself and the processing of the intermediate to the final product. While the properties of the corresponding fractions of the intermediate product already approximately comply with the required specifications when using FT-synthesis [30], the methanol synthesis requires a rather complex product preparation, which involves various processing steps. A possible third option complementing the category of AtJ, the ethanol synthesis, which currently plays a minor role as it is still in an early stage of development, comes with the same disadvantages of high refining effort.

To achieve a simple plant design with minimum complexity, PtL via FT-synthesis is, therefore, the most suitable option. The synthetic fuel production based on a Fischer–Tropsch reactor can mainly be separated into three sections: (I) the generation of synthesis gas (syngas), (II) the generation of synthetic crude (syncrude) via FT-synthesis, and (III) a subsequent separation, upgrading and/or refining of the syncrude to the intermediate or final product [30]. The process chain is described in the following based on this subdivision.

*2.4. Syngas Generation (I)*

The composition of the syngas as feed for the FT-reactor has a significant impact on the synthesis process and is, therefore, largely determined by the desired effects and outputs of this process. Since the desired final product is a mixture of hydrocarbons, the syngas required for a FT process shall mainly consist of the reactants hydrogen ($H_2$) and carbon monoxide (CO).

The various available options for synthesis gas production can be divided into direct syngas production, covering the simultaneous production of both reactants based on a single feedstock, and indirect syngas production, which describes the separate production of the main syngas components, with either the same or different types of feedstock. A direct syngas generation is currently only possible through using bio routes or the StL process. Since biogenic materials or by-products like glycerol are not considered further as raw material in this paper according to the above description, our focus in the subsequent sections will be placed on indirect syngas production.

2.4.1. Hydrogen Production

For the production of hydrogen, the electrochemical process of water electrolysis is used, which splits water into hydrogen and oxygen (1) under electric voltage.

$$H_2O \rightarrow H_2 + \frac{1}{2}O_2 \qquad (1)$$

As of today, there are three main technologies that can be applied. The proton exchange membrane (PEM) and alkaline electrolysis (AEL), both of which have long been used commercially and are technically highly developed, as well as the solid oxide electrolysis cell (SOEC) as a still recent technology [31].

Even if PEM and AEL have proven technology and relatively low specific investment costs, the SOEC comes with some notable advantages. Unlike the others, it is operated with hot steam and not with liquid water, which is why it is alternatively called high temperature electrolysis (HTE). This leads to high electrical efficiencies that already exceed those of the other established technologies by more than 10% [31]. A precondition for operation at the required high temperatures of more than 650 °C is a sufficient heat supply.

Since the FT-reaction is highly exothermic [30], the SOEC is particularly suitable, as the reactors waste heat can be used process internally as heat source for the hot steam generation. An additional improvement of the SOEC is provided by current research activities, which aim at the further development of the co-electrolysis (co-SOEC) providing syngas directly from water, $CO_2$, and electrical energy (2).

$$H_2O + CO_2 \rightarrow H_2 + CO + O_2 \qquad (2)$$

A first small-scale unit of this type has already gone into operation [32].

2.4.2. CO Production

Given that the extraction of carbon monoxide on the basis of industrial residues is excluded for decentralized application, a detour via carbon dioxide has to be taken, as CO does not occur as naturally accessible in the surroundings. Therefore, a carbon capture and utilization (CCU) technology can be applied in which $CO_2$ is extracted from the ambient air via direct air capture (DAC).

Even though different DAC absorption technologies are already under investigation and have been in application for several decades, the adsorption processes are becoming increasingly important for $CO_2$ capture due to their lower specific energy consumption [33]. As one of the possible adsorption types, the temperature-vacuum swing adsorption (TVS) represents a technology that is already in commercial use [34] and qualifies, in particular, for a decentralized PtL process. As with the SOEC, both electrical and thermal energy must

be provided for the operation of a TVS-DAC. The comparatively low required temperature of below 100 °C [35] can also be decoupled from the exothermal synthesis process.

If the above-mentioned co-SOEC is not applied in order to use $CO_2$ directly for synthesis gas production, the conversion of the $CO_2$ into CO must be carried out in a further process step, the reverse water-gas shift reaction (rWGS).

The rWGS describes the hydrogenation of $CO_2$ into CO and $H_2O$ (3).

$$CO_2 + H_2 \leftrightarrow CO + H_2O \ \Delta H_{298K}^0 = +41 \ kJ/mol \tag{3}$$

Since the reactivity of $CO_2$ is lower than that of CO, the chemical equilibrium is on the side of the reactants [36]. A shift of the equilibrium of the endothermic reaction can be obtained by an increase of the reaction temperature. To avoid undesirable side reactions, like methanation and the Sabatier reaction leading to the formation of methane ($CH_4$), reaction temperatures above 700 °C are necessary; to prevent the formation of soot, temperatures above 800 °C should be targeted even at standard conditions [37]. Enhancements of the $CO_2$-conversion and the reduction of the necessary reaction temperature and probability of side reactions can be further achieved by altered pressure, adaption of the input gas shares, or the integration of catalysts [38].

### 2.5. Syncrude Generation (II)

The production of syncrude by the FT synthesis represents the core of the process, as it is decisive for both the syngas composition and, therewith, the upstream process steps as well as the syncrude composition and, therewith, the upgrading requirements of the downstream processes. The FT synthesis refers to a process synthesizing a gas to a synthetic crude oil—the syncrude, composed of a wide range of hydrocarbon chains of different lengths [30]. For a main classification, the reaction is categorized according to the temperature and catalyst type into the iron-based high-temperature FT (Fe-HTFT), iron-based low-temperature FT (Fe-LTFT), and cobalt-based low-temperature FT (Co-LTFT).

The main effect of the different temperature level is a shift in the average syncrude chain lengths. HTFT, due to an increased hydrogenation rate and desorption activity from the catalyst surface, mainly leads to shorter carbon chains <$C_{10}$, whereas LTFT mainly leads to longer carbon chains >$C_{10}$ (mass %) [30]. The major impact of the choice of the catalyst material concerns the share of the syncrude compound classes, which are primarily represented by paraffins (alkanes), olefins (alkenes), aromatics, and oxygenates. With regard to longer-chain hydrocarbons, cobalt-based catalysts lead almost exclusively to the formation of paraffins, whereas, with iron-based catalysts, both olefins and oxygenates are formed in notable proportions. The formation of aromatics is only promoted in the case of Fe-HTFT for certain chain length ranges [30].

As olefins have a deleterious effect on the fuel stability and the proportion of oxygenates should be reduced to a minimum to avoid gum formation [39], a Co-LTFT reactor was selected as appropriate synthesis process step for this study under the consideration of a reduced necessary refining effort. The corresponding mainly occurring chemical reaction equations for the synthesis are:

$$\text{Paraffins}: \ nCO + (2n+1)H_2 \rightarrow H_2(CH_2)_n + nH_2O, \tag{4}$$

$$\text{Olefins}: \ nCO + 2nH_2 \rightarrow (CH_2)_n + nH_2O. \tag{5}$$

### 2.6. Separation, Upgrading, and Refining (III)

The LTFT process provides a syncrude with a temperature around 200 °C, whereby a stepwise subsequent cooling with a cascaded sequence of flash drums is appropriate for the syncrude fractionation. Since the production of syncrude via a FT reactor comes with a broad spectrum of hydrocarbon chains of varying lengths and only the small fraction between $C_8$ and $C_{16}$ is relevant for conventional jet fuel [40], there are always by-products that cannot contribute to the main product. In optimized refineries, those by-products

usually get upgraded or refined to shift them into another chain length range or to provide a set of various final products [39].

This increases the overall plant efficiency significantly. For a decentralized demand-driven production of kerosene, the focus lies on the main product and, thus, no extensive refining for further products was carried out in this study. Hydrocracking was considered as the only refining step focusing on an easily achievable kerosene yield with low process complexity.

Hydrocracking

One objective of a hydrocracking unit in a FT process is to crack heavy long-chain hydrocarbons above the kerosene range into lighter short-chain hydrocarbons and to remove heteroatoms by saturating the compounds via hydrogenation to obtain a paraffinic product [41]. Under the presence of a catalytic material—for an FT feed usually based on palladium or platinum—the syncrude is mixed with hydrogen in the hydrocracking unit at around 360 °C [41].

In terms of the desired fuel properties, the resulting cracked hydrocarbon chains benefit from a further effect that takes place during the hydrocracking. The FT syncrude consists mainly of linear paraffins with a relatively high freezing point, and the hydrocracked output, on the other hand, shows a significant increase of branched iso-paraffins [42], which are necessary to meet the low freezing points of −47 °C for Jet A-1 [18] as defined by the specification (respectively −40 °C for Jet A) [17].

Although still in the development stage and, therefore, not considered in the process outlined in this paper, the integration of hydrocracking in the FT reactor may, in the future, offer a further opportunity to reduce the plant complexity and enhance the process efficiency [14].

### 2.7. Process Recycles

In addition to the main system components, the process-internal gas recycling plays an important role in the process efficiency, as unconverted syngas is recycled after the product separation and fed back into the process, which is described as closed gas loop [30]. The closed gas loop can include an internal recycle, defined here as the recirculation of the tail gases into the syngas in front of the FT reactor and an external recycle describing the recirculation to an earlier process step, for example the rWGS reactor.

### 2.8. Reactor Modelling of Fischer–Tropsch

To assess and evaluate the influence of the FT reactor on the up- and downstream processes, a kinetic Python model of a cobalt-based LTFT reactor was developed based on the current literature.

### 2.8.1. Components and Physical Correlations

To take the wide product range of a FT syncrude into account, n-paraffins from $C_1H_4$ to $C_{45}H_{92}$ and olefins from $C_2H_4$ to $C_{45}H_{90}$ were considered in addition to the main components $H_2$, $CO$, $H_2O$, and $CO_2$. The thermophysical properties of paraffins and olefins are based on [43]. For simplification, we assumed that the saturated hydrocarbons of the FT syncrude consist only of linear paraffins. For olefins, the mean values of linear compounds and isomers with single branching were calculated.

For the chemical components, real gas behavior was taken into account, and the physical behavior was calculated according to the Peng–Robinson equation of state (PREOS) [44].

### 2.8.2. Carbon Number Distribution

The carbon number distribution, which describes the proportions of the individual chain lengths in the product spectrum, was calculated using the Anderson–Schulz–Flory (ASF) distribution [45] according to Equation (6). This sets the product molar fraction $M_n$ of

an individual chain length $n$ in relation to the chain growth probability $\alpha$ (CGP), the value that describes the probability that chain propagation occurs as opposed to chain termination.

$$M_n = \alpha^{(n-1)}(1 - \alpha) \tag{6}$$

The calculation of $\alpha$ is performed by Equation (7) according to [46]:

$$\alpha = \frac{1}{1 + k_a \left( \frac{c_{H_2}}{c_{CO}} \right)^\beta exp \left( \frac{\Delta E_{a,\alpha}}{R} \left( \frac{1}{493.15} - \frac{1}{T} \right) \right)} \tag{7}$$

with $k_a$ as the quotient of rate constants for chain growth termination and propagation, $c_x$ as the molar concentration of species $x$ ($H_2$ and CO), $\beta$ as the syngas ratio power constant, $\Delta E_{a,\alpha}$ as the difference in the activation energy for the termination and propagation reactions of the chain growth mechanism [46], $R$ as the universal gas constant, and $T$ as the temperature. The corresponding values are shown in Table 1.

This definition of $\alpha$ creates a dependency of the growth probability on the reactor temperature and on the syngas composition.

To account for the formation of olefins, which primarily affects the shorter product fractions, a chain length dependent paraffin to olefin ratio was calculated according to [47]:

$$\frac{M_{Olefin}}{M_{Paraffin}} = exp(-d) \tag{8}$$

where the constant $d = 0.3$.

**Table 1.** The kinetic parameter values for the calculation of the chain growth probability (CGP) and reaction rates.

| Parameter | Value | Unit |
|---|---|---|
| CGP [46] | | |
| $k_a$ | 0.0567 | - |
| $\beta$ | 1.76 | - |
| $\Delta E_{a,\alpha}$ | $120.4 \times 10^3$ | J/mol |
| Reaction rate methane [48] | | |
| $a_{CH_4}$ | $-0.86$ | - |
| $b_{CH_4}$ | 1.32 | - |
| $q_{CH_4}$ | 0.46 | - |
| $k_{0,CH_4}$ | $2.925 \times 10^{-7}$ | mol/g/s/MPa$^{(a+b)}$ |
| $E_{a,CH_4}$ | $136 \times 10^3$ | J/mol |
| Reaction rate FT [49] | | |
| $a_{FT}$ | $-0.31$ | - |
| $b_{FT}$ | 0.88 | - |
| $q_{FT}$ | $-0.24$ | - |
| $k_{0,FT}$ | $3.694 \times 10^{-6}$ | mol/g/s/MPa$^{(a+b)}$ |
| $E_{a,FT}$ | $104 \times 10^3$ | J/mol |

### 2.8.3. Reaction Rates

The reaction rate equations and kinetic parameters are based on the works of [48,49]. For reasons as yet unknown in detail, the FT reaction does not fully follow the ASF distribution but shows some deviations for certain chain lengths—especially a significant increase of $C_1$ and a minor decrease of $C_2$ selectivity [30]. For simplification, the decrease of $C_2H_6$ and $C_2H_4$ selectivity was not considered.

The increased fraction of $CH_4$ was taken into account using the reaction rate Equation (9) based on [48] with parameter values according to Table 1:

$$r_{CH_4} = \frac{k_{CH_4} p_{CO}{}^{a_{CH_4}} p_{H_2}{}^{b_{CH_4}}}{\left(1 + q_{CH_4} \frac{p_{H_2O}}{p_{H_2}}\right)^2} \tag{9}$$

with $r_{CH_4}$ as the methane reaction rate (mol/s/g), $a_{CH_4}$ and $b_{CH_4}$ as the reaction orders of the partial pressures $p$ for CO and $H_2$, and $q_{CH_4}$ as the water effect constant for $CH_4$ formation.

The temperature dependent reaction rate constant $k_{CH4}$ is defined as

$$k_{CH_4} = k_{0,CH_4} exp\left(\frac{E_{a,CH_4}}{R}\left(\frac{1}{493.15} - \frac{1}{T}\right)\right) \tag{10}$$

where $k_{0,CH_4}$ describes the reaction rate constant at 493.15 K and $E_{a,CH_4}$ the activation energy for $CH_4$ formation [48]. The special feature of this equation compared to other kinetic datasets available in the current literature is the consideration of the influence of water, which is one main by-products of the process and may as well be present in the syngas feed.

For the further product spectrum, the reaction rate is defined based on [49] by the equation:

$$r_{FT} = \frac{k_{FT} p_{CO}{}^{a_{FT}} p_{H_2}{}^{b_{FT}}}{\left(1 + q_{FT} \frac{p_{H_2O}}{p_{H_2}}\right)} \tag{11}$$

with $r_{FT}$ as the FT reaction rate, $k_{FT}$ as temperature dependent reaction rate constant calculated via

$$k_{FT} = k_{0,FT} \exp\left(\frac{E_{a,FT}}{R}\left(\frac{1}{493.15} - \frac{1}{T}\right)\right) \tag{12}$$

where $k_{0,FT}$ is the reaction rate constant at 493.15 K, and $E_{a,FT}$ is the activation energy. Both the reaction rates presented above refer to the molar amount of reacted CO molecules.

The chain length specific reaction rates $r_{i,n}$ for all considered chain lengths $n$ and $i \in$ paraffin $\wedge$ olefin for all following equations were calculated via

$$r_{i,n} = \frac{M_{i,n} m_{mol,i,n}}{\sum_{n=1}^{45} M_{i,n} m_{mol,i,n}} r_{FT} \tag{13}$$

with $m_{mol}$ as the molar mass (g/mol) where—to account for the methane deviation—all $r_{i,n}$ for n > 1 are multiplied with the FT methane reaction rate according (14), and $r_{i,n}$ is substituted for $i$ = paraffin and $n$ = 1 (15):

$$r_{i,n} = r_{i,n}\left(1 + \frac{r_{paraffin,1}}{r_{FT}}\right), \tag{14}$$

$$r_{paraffin,1} = r_{CH_4}. \tag{15}$$

A normalized dimensionless selectivity $S$ is introduced (16), which is used to redefine the specific reaction rates (17) to meet the reaction rate $r_{FT}$ of the total synthesis process:

$$S_{i,n} = \frac{r_{i,n}}{\sum_{n=1}^{45} r_{i,n}}, \tag{16}$$

$$r_{i,n} = S_{i,n} r_{FT}. \tag{17}$$

### 2.8.4. Partial Pressures

The partial pressures of the relevant components that form the basis for the above reaction rate Equations (9) and (11) were calculated based on the $H_2$ usage ratio $ur_{H_2}$ and the reaction rates showing the iterative character of the reactor calculation.



The resulting pressures were either calculated as the mean values between the partial pressures at the reactor outlet after the reaction and the reactor inlet for reactor types with a plug flow reactor characteristic (PFR) or calculated based entirely on the product composition at the reactor outlet according to Equations (18)–(20) for reactor types that can be classified as a continuous stirred-tank reactor (CSTR):

$$p_{H_2,out} = p\left(\frac{c_{H_2,in}\dot{N}_{total} - 2m_{cat}ur_{H_2}\sum_{n=1}^{45} r_{i,n}}{\dot{N}_{total}}\right) \tag{18}$$

$$p_{CO,out} = p\left(\frac{c_{CO,in}\dot{N}_{total} - 2m_{cat}\sum_{n=1}^{45} r_{i,n}}{\dot{N}_{total}}\right) \tag{19}$$

$$p_{H_2O,out} = p\left(\frac{c_{H_2O,in}\dot{N}_{total} + 3m_{cat}\sum_{n=1}^{45} r_{i,n}}{\dot{N}_{total}}\right) \tag{20}$$

with $\dot{N}_{total}$ as the syngas particle flow (mol/s) and $m_{cat}$ as the reactor catalyst mass (g).

### 2.8.5. $H_2$ Usage Ratio

The $H_2$ usage ratio describes the ratio between the converted $H_2$ and converted CO during the FT reaction. The specific proportion of $ur_{H_2i,n}$ varies for each paraffin of different length according to Equation (4) between 2 and 3. The $H_2$ usage ratio for olefins equals 2 over the whole product range as per Equation (5). The average $H_2$ usage ratio of the entire product spectrum was, therewith, calculated based on the component type and chain length specific reaction rate shares:

$$ur_{H_2} = \sum_{n_1=1}^{45} \frac{r_{i_1,n_1}}{\sum_{n_2=1}^{45} r_{i_2,n_2}} ur_{H_2i_1,n_1}. \tag{21}$$

### 2.8.6. CO Conversion

The CO conversion $u_{CO}$ as one key parameter of the reactor operation was calculated inter alia based on the above-described reaction rates and the gas hourly space velocity (GHSV), which indicates the hourly syngas flow rate per gram of catalyst loading:

$$u_{CO} = \frac{2\frac{\dot{V}_{total}}{GHSV}\sum_{n=1}^{45} r_{i,n}}{c_{CO,in}\dot{N}_{total}} \tag{22}$$

with $\dot{V}_{total}$ as the volumetric syngas flow (NL/h).

### 2.8.7. Calculation Method

Figure 3 demonstrates the simplified calculation flow sheet, which is based on the equations presented above. The interdependencies between the partial pressures, the reaction rates and the $H_2$ usage ratio led to an iterative calculation that was performed until a specific deviation for the CO conversion and the $H_2$ usage ratio being below 0.001%. The predefined input compound that is fed to the reactor as syngas is defined in particular by the shares of its components (providing the necessary molar concentrations of $H_2$, CO and $H_2O$), the temperature, the pressure, and the flow rate.

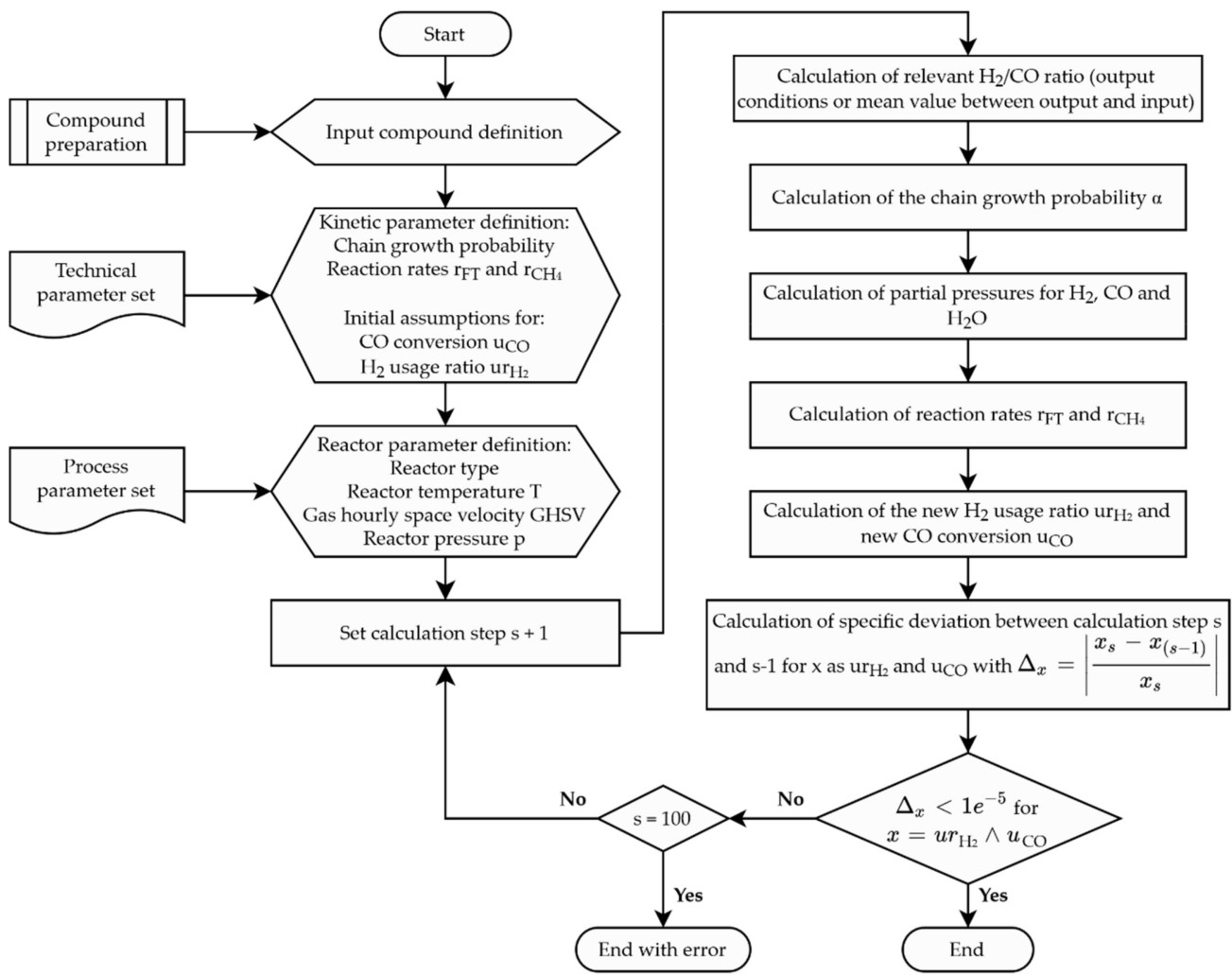

**Figure 3.** Flow chart of the Fischer–Tropsch (FT) reactor calculation model.

*2.9. Model Validation*

A comparison of the experimental [48,49] and modeled reaction rates, showing the expected reactor behavior and leading to satisfactory coefficients of determination is shown in Appendix A.

For a further evaluation of the model validity with regard to the intended model purpose of representing the possible degrees of freedom by means of conclusive parameter correlations, the main dependencies between the adjustments of the key reactor parameters and the reactor behavior were examined. The main correlations for cobalt-based FT-reaction in consideration of the model simplifications made are shown in Table 2.

**Table 2.** Dependencies of the reactor behavior on the main Fischer–Tropsch reactor operation parameters [30,46,50,51] for cobalt-based catalysts.

| | Operation Parameters: | | | |
|---|---|---|---|---|
| | ▲ Temperature | ▲ Pressure | ▲ GHSV | ▲ $H_2$/CO ratio |
| Reactor behavior: | | | | |
| CGP | ▼ | ● | ● | ▼ |
| $CH_4$ selectivity | ▲ | ▼ | ▲ | ▲ |
| Syngas conversion | ▲ | ▲ | ▼ | ○ |

▲ = increase, ▼ = decrease, ● = minor impact, ○ = more complex dependency.

The presented correlations are shared by a majority of the available literature concerning the cobalt-based FT reaction. Variations can be found regarding the impact of a pressure increase on the chain growth potential and the effect of a change in GHSV on the $CH_4$ selectivity. While the author of [30] concluded a rising CGP with an increase of the reactor pressure, other studies tended to conclude that the pressure impact on the CGP was negligible [46,50].

A decrease of the methane selectivity resulting from an increase in GHSV, which was concluded in [30], is opposed to an increase of $CH_4$ selectivity [48,51] for cobalt-based catalysts at the typical FT operation parameter ranges, which can be attributed to the influence of the by-product water, which suppresses methane formation [48,51] and has a share that is increased at elevated syngas conversion rates. One reason for the varying dependencies in different sources can be the significant influence of the catalyst material on the reactor behavior, thus, in the following, the dependencies that clearly address cobalt-based catalysts are assessed for model validation.

Figure 4 shows the effects on the reactor for varying the temperature (a) and pressure (b) based on the reactor model. The expected reactor behavior is well represented by the model results. The same applies for the impact of the GHSV (c) and the syngas reactants ratio (d).

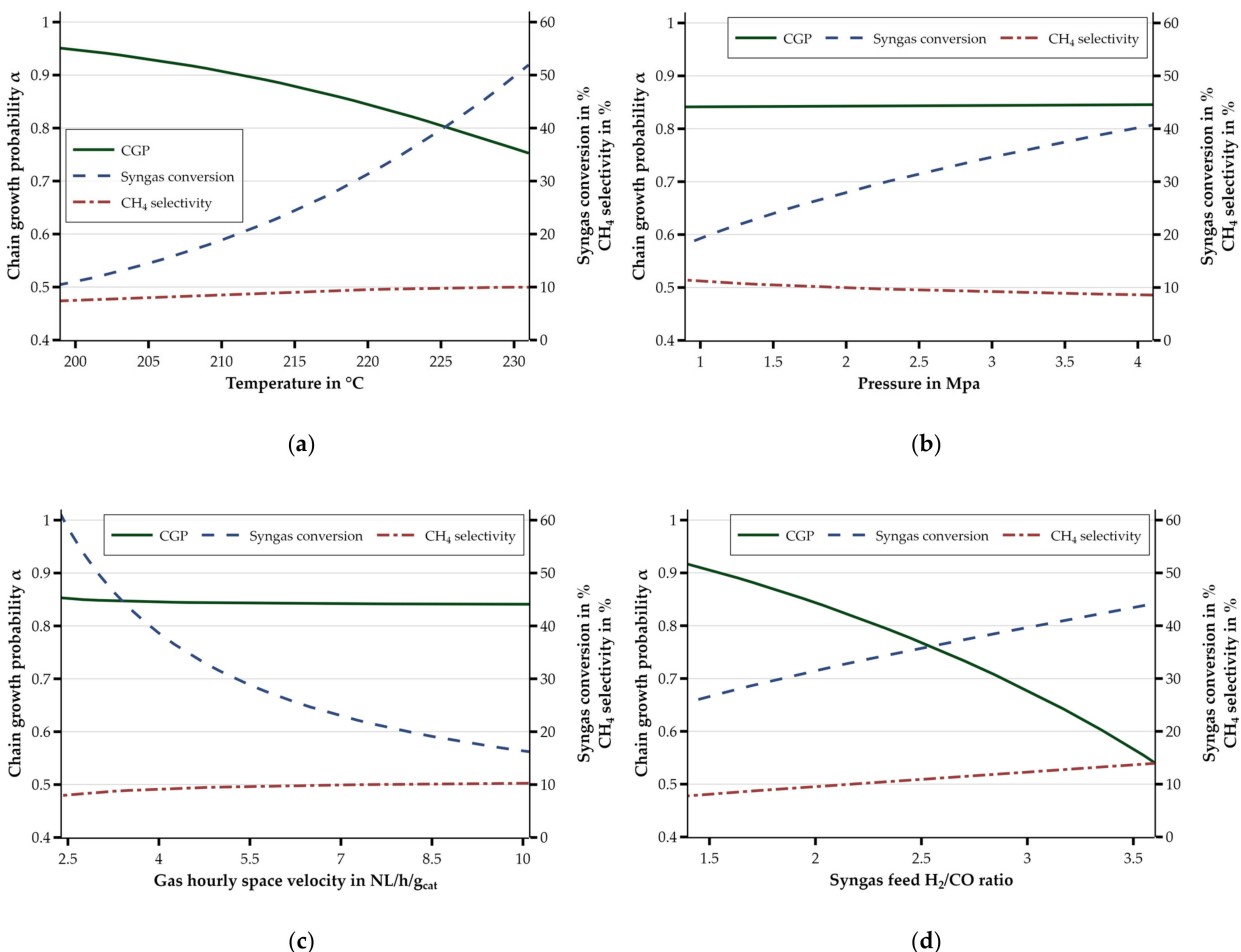

**Figure 4.** Key parameters of the FT reactor calculated on the basis of the model showing (**a**) the temperature-dependency for a fixed-bed-reactor with p = 2.5 MPa, gas hourly space velocity (GHSV) = 5 NL/h/$g_{cat}$, and H2/CO syngas feed ratio = 2; (**b**) the pressure-dependency for a fixed bed reactor with T = 220 °C, GHSV = 5 NL/h/$g_{cat}$, and H2/CO syngas feed ratio = 2; (**c**) the dependency from the GHSV for a fixed-bed-reactor with T = 220 °C, p = 2.5 MPa, and H2/CO syngas feed ratio = 2; and (**d**) the dependency on the H2/CO syngas feed ratio for a fixed bed reactor with T = 220 °C, p = 2.5 MPa, and GHSV = 5 NL/h/$g_{cat}$.

## 3. Results and Discussion

To show the relevance of the FT reactor and its operation parameters as part of a PtL process regarding the following downstream and previous upstream processes, a closer look at the resulting syncrude composition is necessary. Figure 5 visualizes the impacts of a pressure adjustment (a) and an increase of the GHSV (b) on the converted syngas showing the selectivity and its share of hydrocarbon chains with chain lengths between $C_8$ and $C_{16}$ representing the compounds that are considered part of kerosene.

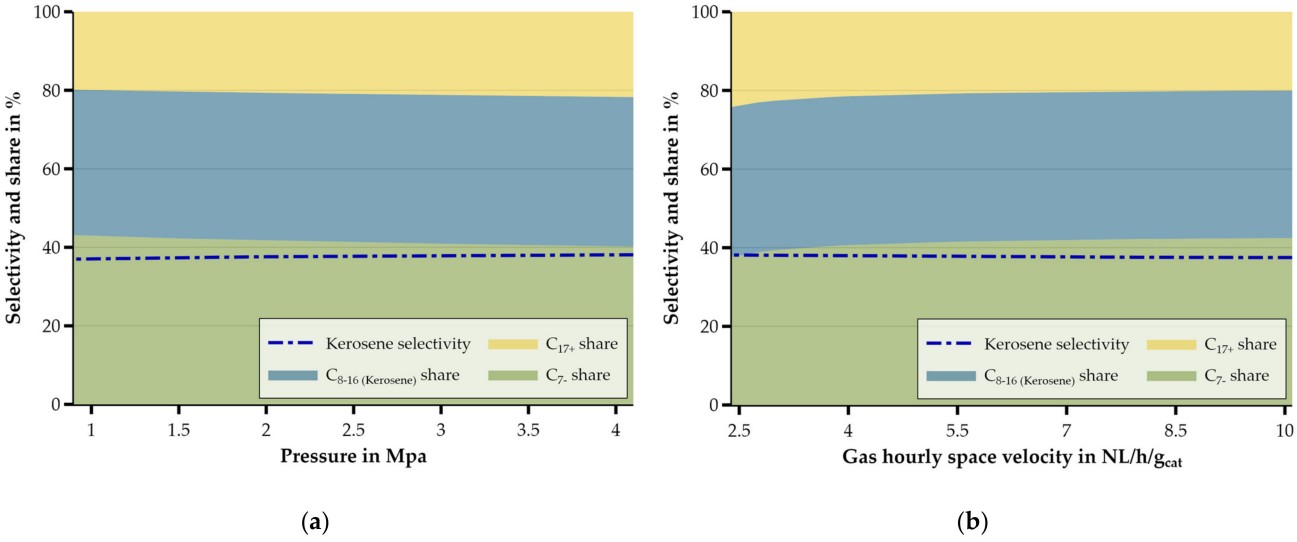

(a)                                                                                    (b)

**Figure 5.** Reactor selectivity shares of different carbon chain length ranges and the kerosene range selectivity on the basis of the model showing (**a**) the pressure-dependency for a fixed bed reactor with T = 220 °C, GHSV = 5 NL/h/$g_{cat}$, and H2/CO syngas feed ratio = 2; and (**b**) the dependency from the GHSV for a fixed-bed-reactor with T = 220 °C, *p* = 2.5 MPa, and H2/CO syngas feed ratio = 2.

Additionally, the shares for the shorter hydrocarbon chain length ranges $C_{7-}$ and longer ranges $C_{17+}$ are provided. As can be expected from Figure 4b,c due to the insignificant impact on the CGP, adjustments of those two operation parameters only show minor effects on the syncrude composition. The biggest impacts concerning the reactor reactivity resulting at elevated pressures and the decrease of syngas conversion with increased GHSV mainly address the basic plant design in terms of the size and structure. Thus, they are not parameters governing the surrounding process steps but are part of an overarching dimensioning and plant optimization, in particular with regard to deactivation and the lifetime of the reactor catalyst load [52].

The effects of variations of the reactor temperature are provided in Figure 6, showing a strong influence on the syncrude distribution. According Figure 6, the decreasing probability of the chain growth caused by an increase of the temperature led to high shares of light short-chained hydrocarbons at elevated temperatures. The selectivity maximum for hydrocarbons within the kerosene chain length range was found at around 220 °C with close to 38%, and, although the focus of this work and the process under consideration is on kerosene as the main product, maximizing the straight run kerosene output of the reactor may not be expedient in consideration of the overall process efficiency.

As the refining of the syncrude is crucial to meet relevant fuel specifications (which is why a hydrocracking unit was selected as part of the process), the aim should be to maximize the share of syncrude fractions that can further be refined to on-specification products. In the case of a synthesis with a subsequent hydrocracker where long-chained hydrocarbons can be cracked into the kerosene range, optimizing the yield of hydrocarbons with a chain length above $C_8$ might prove beneficial.

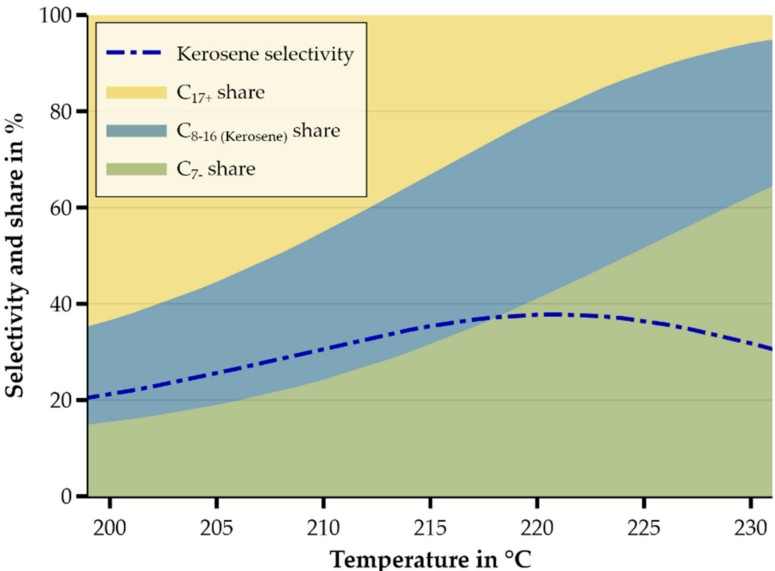

**Figure 6.** The temperature dependency on selectivity shares and the kerosene range selectivity for a fixed bed reactor with $p$ = 2.5 MPa, GHSV = 5 NL/h/g$_{cat}$, and H$_2$/CO syngas feed ratio = 2.

However, certain reactor conditions that would lead to unwanted side effects, like an increased reactor deactivation caused by high syngas conversion rates or a low reactivity resulting in large unit sizes to achieve the desired product quantities, should be avoided. The design of the main operation point should, thus, be based on a complex techno-economical optimization to achieve the best trade-off between the relevant operation parameters.

A similar influence as with the temperature dependence was also observed with the dependence on the H$_2$/CO ratio of the syngas feed shown in Figure 7. With the increasing proportion of hydrogen, the CGP decreased and, thus, the share of the light hydrocarbons increased; however, in contrast to the reactor temperature, the setting of the H$_2$/CO ratio was subject to certain requirements to ensure a uniform process operation, which results from the reactor operation itself.

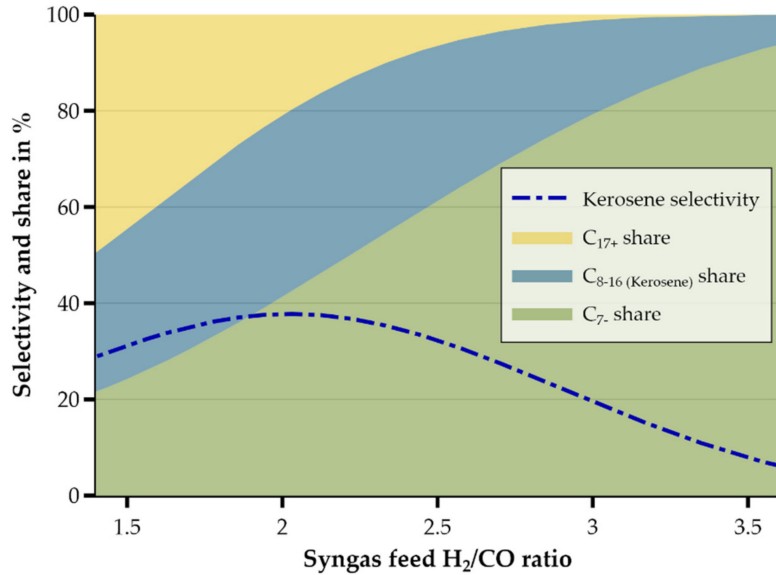

**Figure 7.** The syngas reactants ratio dependency on the selectivity shares and kerosene range selectivity for a fixed bed reactor with T = 220 °C, $p$ = 2.5 MPa, and GHSV = 5 NL/h/g$_{cat}$.

One common aim in the operation of FT reactors is to meet the reactors $H_2/CO$ usage ratio with the $H_2/CO$ syngas feed ratio [30] in order to maintain a constant proportion between the reactants throughout the entire reaction. Accordingly, even though the $H_2/CO$ feed ratio has a relevant impact on the syncrude distribution and the reactor activity, its use as a controlling parameter is strongly limited in favor of a homogeneous synthesis process.

Although the reactor temperature has primarily emerged as an authoritative control parameter that comes along with a high degree of freedom, the previous evaluations show the decisive possibilities of influencing the syncrude composition and general reactor behavior by means of the key operation parameters. The impact on the upstream processes of the syngas generation resulted primarily from the feed gas composition, which is required for stable reactor operation at the desired operating point. The downstream processes, which include, in particular, the product separation and refining, were mainly affected by the reactor selectivity, as their specific design should be based on the composition of the supplied syncrude.

## 4. Conclusions

The presented work describes the process chain of sustainable decentralized production of kerosene based on renewable energy and current technologies. According to the general criteria that have to be met for a decentralized fuel supply, the power-to-liquid process was selected as the preferred option due to its flexible energy supply possibilities and minor impacts on land and water use in comparison to biomass-to-liquid processes.

As one of two possible process routes for a kerosene production based solely on electrical energy, synthesis via Fischer–Tropsch reaction was preferred to the other variant due to its simple process structure and low necessary refining effort to obtain kerosene. Relevant options for syngas generation were sketched, and hydrocracking was selected as the only refining step for upgrading to an on-specification product. Based on a developed Python model for the simulation of a FT reactor, the effects of the main operation parameters on the product selectivity and reactor activity are provided to show the key position of the FT reactor as a governing process in the production chain on the one hand, and its flexibility toward a targeted syncrude production on the other.

The dependencies of the operation parameters and product composition on the reactor behavior represent the complex interrelationships in a FT reactor. The product selectivity, which is highly determined by the reactor temperature, was decisive for the downstream processes of product separation and hydrocracking and the upstream processes due to its influence on the $H_2/CO$ usage ratio, which should be met by the $H_2/CO$ syngas feed ratio.

The reactor temperature not only played a major role regarding the chain length distribution of the syncrude but also significantly affected the reactor activity leading to increased conversion rates at elevated temperatures. The influence of the reactor pressure and the GHSV on the product selectivity and reaction activity was negligible. Those parameters can, therefore, be influenced by the overarching plant concept and are not primarily governed by the FT process. However, as they both have a relevant impact on the syngas conversion whose range is crucial to meet a compromise between the catalyst deactivation caused by increased water shares and the process efficiency, which is negatively affected by increasing amounts of unconverted syngas, a favorable reactor operation should be aimed for.

As a summary, the presented reactor parameters and the reactor temperature in particular offer a high potential for a targeted process operation, and the dependencies showed the decisive role of the Fischer–Tropsch reactor for the PtL process. However, as the FT reactor is also part of the overall system in the PtL process, a technical and/or economical process optimization toward a maximal process efficiency can only be performed under consideration of all the relevant process steps.

## 5. Outlook

Future work will extend the Python model with additional process steps as well as internal and external gas recycling and relevant economic and operational parameters to depict the entire PtL production path from energy supply to the final product. On this basis, both a process optimization for the decentralized production of kerosene and a system-analytical assessment of the future role of a sustainable kerosene production can be performed.

**Author Contributions:** Conceptualization, J.K. and A.M.; methodology, A.M.; software, A.M.; validation, A.M.; investigation, A.M.; writing—original draft preparation, A.M.; writing—review and editing, A.M. and J.K.; visualization, A.M.; supervision, J.K.; project administration, J.K.; funding acquisition, J.K. All authors have read and agreed to the published version of the manuscript.

**Funding:** This research was funded by the German Federal Ministry for the Environment, Nature Conservation and Nuclear Safety via the project Klimaneutrale Alternative Kraftstoffe (ProQR) together with the Deutsche Gesellschaft für Internationale Zusammenarbeit (GIZ).

**Conflicts of Interest:** The authors declare no conflict of interest.

## Abbreviations

| Parameter | Description |
|---|---|
| AEL | Alkaline electrolysis |
| ASF | Anderson–Schulz–Flory |
| AtJ | Alcohol-to-jet |
| BtL | Biomass-to-Liquid |
| CCU | Carbon capture usage |
| CGP | Chain growth probability [-] |
| Co-LTFT | Cobalt based low temperature Fischer–Tropsch |
| CSTR | Continuous stirred-tank reactor |
| DAC | Direct air capture |
| exp | Experimental |
| Fe-HTFT | Iron based high temperature Fischer–Tropsch |
| Fe-LTFT | Iron based low temperature Fischer–Tropsch |
| FT | Fischer–Tropsch |
| FT-SPK | Fischer–Tropsch synthetic paraffinic kerosene |
| HTE | High temperature electrolysis |
| PEM | Proton exchange membrane |
| PBtL | Power-Biomass-to-Liquid |
| PFR | Plug flow reactor |
| PREOS | Peng–Robinson equation of state |
| PtL | Power-to-Liquid |
| rWGS | Reverse water-gas shift |
| SAF | Sustainable aviation fuel |
| SOEC | Solid oxide electrolysis cell |
| SPK | Synthetic paraffinic kerosene |
| StL | Sun-to-Liquid |
| TVS | Temperature–vacuum swing adsorption |
| $a$ | Reaction order of partial pressure CO [-] |
| $a_0$ | Adsorption coefficient at 493.15 K [-] |
| $b$ | Reaction order of partial pressure $H_2$ [-] |
| $c$ | Concentration [%] |
| $d$ | Paraffin to olefin ratio constant [-] |
| $E_a$ | Activation energy [J/mol] |
| err | Error [%] |
| GHSV | Gas hourly space velocity [NL/(s $g_{catalyst}$)] |

| | |
|---|---|
| $k_0$ | Reaction rate constant at 493.15 K [-] |
| $k_a$ | Rate constant of ratio of termination and propagation [-] |
| $k_{CH4}$ | Temperature dependent reaction rate constant of methane [-] |
| $k_{FT}$ | Temperature dependent reaction rate constant of FT products [-] |
| M | Molar fraction [-] |
| $m_{cat}$ | Catalyst mass [g] |
| $m_{mol}$ | Molar mass [g/mol] |
| $\dot{N}$ | Particle flow [mol/s] |
| n | Carbon number [-] |
| p | Pressure [MPa] |
| q | Water effect constant [-] |
| R | Universal gas constant [(kg m$^2$)/(s$^2$ mol K)] |
| $R^2$ | Coefficient of determination [-] |
| r | Reaction rate [mol/(s g$_{catalyst}$)] |
| S | Selectivity [-] |
| T | Temperature [K] |
| t | Temperature [°C] |
| u | Conversion [-] |
| ur | Usage ratio [-] |
| $\dot{V}$ | Volume flow [NL/h] |
| α | Chain growth probability [-] |
| β | Syngas ratio power constant [-] |
| Δ | Difference [-] |

## Appendix A  Reaction Rate Comparison

Table A1 shows the experimental derived reaction rates by [48,49] for various reactor conditions with variations in the syngas composition, system pressure, and space velocity as well as the corresponding modeled reaction rates.

**Table A1.** Comparison of the experimental [48,49] and modeled reaction rates.

| Run | $p_{CO}$ [MPa] | $p_{H_2}$ [MPa] | GHSV [NL/g/h] | $r_{FT,exp}$ [mol/g/h] | $r_{CH_4,exp}$ [mol/g/h] | $r_{FT,model}$ [mol/g/h] | $r_{CH_4,model}$ [mol/g/h] [a] | $err_{FT}$ [%] [b] | $err_{CH_4}$ [%] [b] |
|---|---|---|---|---|---|---|---|---|---|
| 2 | 0.710 | 1.420 | 16.0 | 0.0205 | 0.0022 | 0.0193 | 0.0020 | 5.85 | 8.02 |
| 3 | 0.710 | 1.420 | 10.0 | 0.0207 | 0.0021 | 0.0189 | 0.0019 | 8.70 | 12.63 |
| 4 | 0.710 | 1.420 | 6.0 | 0.0204 | 0.0019 | 0.0182 | 0.0017 | 10.93 | 14.86 |
| 5 | 0.710 | 1.420 | 3.0 | 0.0188 | 0.0016 | 0.0166 | 0.0012 | 11.49 | 30.09 |
| 7 | 0.710 | 1.065 | 16.0 | 0.0178 | 0.0015 | 0.0150 | 0.0014 | 15.51 | 6.82 |
| 8 | 0.710 | 1.065 | 10.0 | 0.0151 | 0.0014 | 0.0147 | 0.0013 | 2.72 | 7.14 |
| 9 | 0.710 | 1.065 | 3.0 | 0.0129 | 0.0011 | 0.0129 | 0.0008 | −0.08 | 23.78 |
| 10 | 0.710 | 1.065 | 6.0 | 0.0145 | 0.0013 | 0.0141 | 0.0011 | 2.55 | 10.35 |
| 12 | 0.710 | 0.710 | 16.0 | 0.0100 | 0.0008 | 0.0105 | 0.0008 | −5.50 | 3.23 |
| 13 | 0.710 | 0.710 | 10.0 | 0.0112 | 0.0008 | 0.0103 | 0.0008 | 7.77 | 2.50 |
| 14 | 0.710 | 0.710 | 3.0 | 0.0099 | 0.0006 | 0.0091 | 0.0005 | 7.78 | 12.39 |
| 15 | 0.710 | 0.710 | 6.0 | 0.0111 | 0.0007 | 0.0010 | 0.0007 | 10.36 | 5.46 |
| 17 | 0.487 | 1.217 | 16.0 | 0.0226 | 0.0025 | 0.0192 | 0.0023 | 15.31 | 7.40 |
| 18 | 0.487 | 1.217 | 10.0 | 0.0206 | 0.0025 | 0.0188 | 0.0022 | 8.79 | 11.23 |
| 19 | 0.487 | 1.217 | 3.0 | 0.0199 | 0.0022 | 0.0171 | 0.0015 | 14.23 | 31.99 |
| 20 | 0.487 | 1.217 | 6.0 | 0.0192 | 0.0023 | 0.0182 | 0.0020 | 5.10 | 14.69 |
| 22 | 0.608 | 1.217 | 16.0 | 0.0175 | 0.0021 | 0.0178 | 0.0019 | −1.60 | 9.85 |
| 23 | 0.608 | 1.217 | 10.0 | 0.0172 | 0.0021 | 0.0174 | 0.0018 | −1.10 | 14.46 |
| 24 | 0.608 | 1.217 | 6.0 | 0.0175 | 0.0019 | 0.0168 | 0.0016 | 4.17 | 16.56 |
| 25 | 0.608 | 1.217 | 3.0 | 0.0182 | 0.0016 | 0.0154 | 0.0011 | 15.16 | 27.75 |
| 27 | 0.811 | 1.217 | 16.0 | 0.0157 | 0.0015 | 0.0162 | 0.0015 | −3.06 | 6.33 |
| 28 | 0.811 | 1.217 | 10.0 | 0.0184 | 0.0015 | 0.0158 | 0.0014 | 14.18 | 8.06 |
| 29 | 0.811 | 1.217 | 6.0 | 0.0148 | 0.0013 | 0.0151 | 0.0012 | −2.30 | 11.31 |
| 30 | 0.811 | 1.217 | 3.0 | 0.0141 | 0.0010 | 0.0138 | 0.0008 | 2.48 | 20.13 |

[a] The methane reaction rate was calculated via the provided $CH_4$ selectivity [48]. [b] Error = $(r_{x,exp}-r_{x,model})/r_{x,exp}$.

The measurements were performed at a reactor temperature of 220 °C. As there was no continuous information regarding the total system pressure provided for each calculation run, the model reactor pressure was considered as the sum of the reported hydrogen and carbon monoxide partial pressures. The parity plots of both reaction rates are shown in Figure A1 for the modeled values, and the rates were calculated by Ma et al. [48,49].

With the coefficients of determination of $R^2 = 0.904$ (F = 0.91 with $F_{critical(0.95)} = 3.84$) for the FT reaction rate $r_{FT,model}$ and $R^2 = 0.930$ (F = 1.14 with $F_{critical(0.95)} = 3.84$) for the methane reaction rate $r_{CH_4,model}$, the model provided an adequate representation of the experimental results. The modeled results further showed the expected behavior with changed process parameters. Deviations from the calculated values provided by [48,49] mainly resulted from additionally modeled process-related influences on the chain growth probability, an additional consideration of olefins and the aforementioned uncertain deviation between the reactants partial pressures and the total reactor pressure.

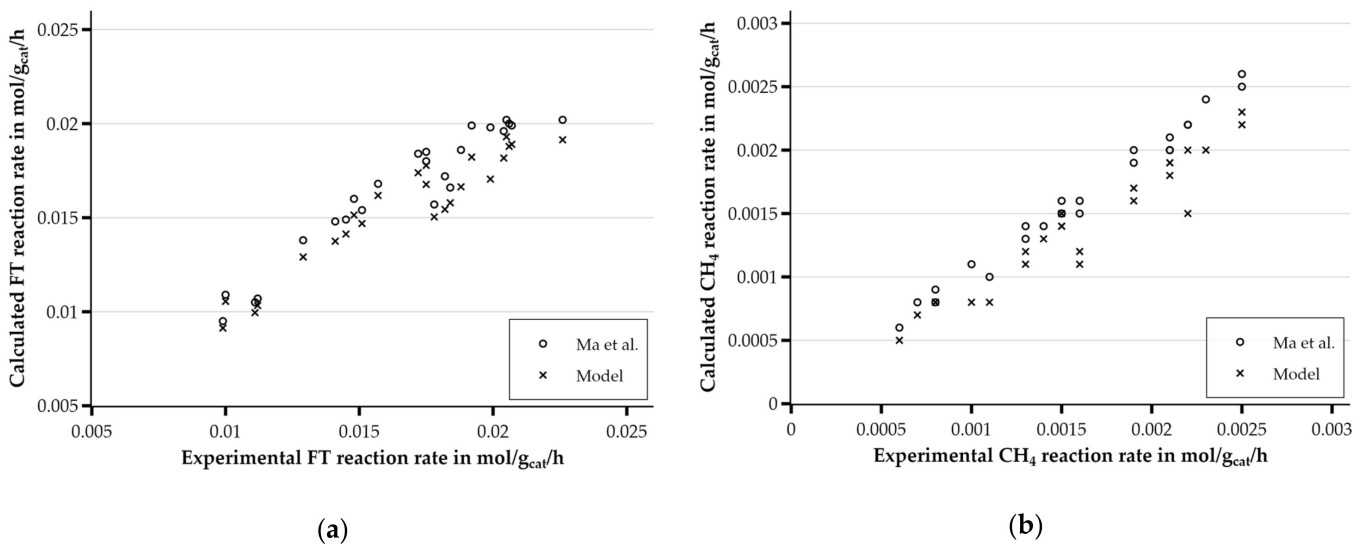

(**a**)  (**b**)

**Figure A1.** Parity plots for the modeled and literature-based [48,49] (**a**) FT reaction rates and (**b**) methane reaction rates.

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
