# Peer review of "Fischer–Tropsch Synthesis as the Key for Decentralized Sustainable Kerosene Production†"

_energies, doi:10.3390/en14071836_

Round 1

Reviewer 1 Report

The authors presented their FT simulation work of sustainable decentralized production of kerosene based on renewable energy and current technologies using a model developed by Pythan. FT product selectivity obtained from model under different process conditions matches with literature. The dependencies show the decisive role of the Fischer-Tropsch reactor for the PtL process. The process optimization performed to meet the goal of maximum conversion. 

The presented data with studied limited process variable although informative but looks incomplete. I would suggest the following points to be considered to further enhance the scientific merits of the paper.

  1. A thorough English usage check is needed. 
  2. It is understood that reaction temperature is a key parameter that controls product selectivity. what about the effects of H2/CO on product selectivity?
  3. When you perform kinetic model based on experimental data, how catalyst deactivation is accounted for different process conditions?
  4. The complete program details may be provided if it is not associated with proprietary.   
  5. What is the significance error of the plots, calculated FT vs experimental FT and calculated CH4rate vs experimental CH4.?
  6. Similar plots made with product Kerosene would be interesting.  

Author Response

Point 1: A thorough English usage check is needed. 

Response 1: An English check was performed by MDPI English Editing and is incorporated in the resubmitted manuscript.

Point 2: It is understood that reaction temperature is a key parameter that controls product selectivity. what about the effects of H2/CO on product selectivity?

Response 2: The impact of an adjusted H2/CO ratio is provided in Figure 7. Although it is shown that the syngas ratio highly affects the product selectivity, it is not considered as one of the key parameters. The reason is that the syngas feed ratio is subject to the reactor usage ratio: If the feed ratio differs from the usage ratio, the reaction conditions over the reactor cannot be uniform. Local deviations might increase the catalyst deactivation or might lead to unexpected or random product selectivity. Thus, for operation of FT reactors, matching the syngas feed ratio to the reactor usage ratio is in general recommended.

The reactor temperature on the other side is not governed by operational characteristics of the FT reactor, but can be selected as the governing parameter.

That is why the feed gas ratio is not, but the temperature is described as key parameter, even though both significantly affect the product selectivity.

Point 3: When you perform kinetic model based on experimental data, how catalyst deactivation is accounted for different process conditions?

Response 3: A catalyst deactivation is currently not considered in the model.

The catalyst deactivation is indeed an interesting and relevant topic for the operation of catalytic reactors. For our current studies, we focused on the initial process setup, which shall guarantee uniform process conditions over a reasonable time period. A basic requirement for the selection of the kinetic reactor parameters was therefore a comprehensible diligence regarding the topic of deactivation. The chosen studies provide parameters which are measured between a stable process period of around 365 to 900 hours of time on stream.

When assessing the reactor operation at varying process conditions based on those kinetic parameters, the range of process parameters in which the experimental data were derived without noticing severe impacts on the catalyst deactivation can be taken as one indicator for reasonable parameter range for this specific kinetic model. Additionally, prominent operating conditions which lead to an increase in reactor deactivation like e.g., excessive shares of water have to be taken into account when evaluating the different process conditions.

Nevertheless, especially for implementation of such a kinetic model in temporally resolved evaluations, the impact of the catalyst deactivation has to be considered and has to be accounted for either by an adjustment of catalyst activity and/or product selectivity, or by taking a frequent catalyst replacement as a basis.

Point 4: The complete program details may be provided if it is not associated with proprietary.

Response 4: The complete program consists of an extensive framework and multiple implemented modules which cannot be used detached, thus, the provision as part of this publication is not practicable. Notwithstanding, the provision of the entire model, including framework and modules is planned as part of the mentioned open-source-strategy, but as this is an extensive process, this will unfortunately take a few more months.

Point 5: What is the significance error of the plots, calculated FT vs experimental FT and calculated CH4rate vs experimental CH4.?

Response 5: With a confidence interval of 95%, the F-value for FT reaction rate is 0.91 and for CH4 reaction rate 1.14, both with a critical F-value of 3.84.

The values were implemented in the Appendix of the manuscript and are part of the resubmitted version.

Point 6: Similar plots made with product Kerosene would be interesting.

Response 6: As the focus of existing studies and experimental analyses is not set on kerosene as desired product, but on other specific questions of the respective studies, we did not find any detailed data regarding the kerosene yields of FT reactor setups.

However, we agree that such comparisons would be valuable and we hope that with the currently increasing interest in the topic of FT syncrude for the aviation sector, such data will be also available in the near future.

Reviewer 2 Report

The Fischer-Tropsch synthesis as key for a decentralized sustainable kerosene production is shown in this paper. The Authors effects of adjustments of the main operation parameters on the reactor behavior processes are described. The reactor model (proposed by Authors) is based on reaction kinetics according current literature. The research results prove the governing influence of the Fischer-Tropsch reactor on the Power-to-Liquid process and show its flexibility regarding the desired product fraction output, which makes it an appropriate solution for a decentralized kerosene production. 
The novelty of this work is the description of a process chain tailored to a decentralized and sustainable production of kerosene which can be used directly on site. Moreover, the manuscript also describes the development of a   reactor model as part of a future open source process simulation model based on Python, showing the importance of the Fischer-Tropsch reactor as core of the Power-to-Liquid process. 
The work shows that the presented reactor parameters offer high potential for a targeted process operation and the dependencies show the decisive role of the Fischer-Tropsch reactor for the PtL process. 
The work is good described.
The references were well matched to the research topic.
It should be good to:
-  increase the resolution of some figures (e.g. Figure 4),
- analyze or rewrite the conclusions.

Author Response

Point 1: increase the resolution of some figures (e.g. Figure 4)

Response 1: The size of the small figures was increased to improve readability. This affects Figure 4 and Figure 5.

Point 2: analyze or rewrite the conclusions.

Response 2: The conclusion section was revised and especially the significant role of the reactor temperature was pointed out additionally.

Round 2

Reviewer 1 Report

The authors response is greatly appreciated. The paper can be published in its current form. 

This manuscript is a resubmission of an earlier submission. The following is a list of the peer review reports and author responses from that submission.